# Cytokine Signaling in Pediatric Kidney Tumor Cell Lines WT-CLS1, WT-3ab and G-401

**DOI:** 10.3390/ijms25042281

**Published:** 2024-02-14

**Authors:** Elizaveta Fasler-Kan, Milan Milošević, Sabrina Ruggiero, Nijas Aliu, Dietmar Cholewa, Frank-Martin Häcker, Gabriela Dekany, Andreas Bartenstein, Steffen M. Berger

**Affiliations:** 1Department of Pediatric Surgery, Children’s Hospital, Inselspital Bern, University of Bern, CH-3010 Bern, Switzerland; milan.milosevic@insel.ch (M.M.); sabrina.ruggiero@insel.ch (S.R.); dietmar.cholewa@insel.ch (D.C.); gabriela.dekany@insel.ch (G.D.); andreas.bartenstein@insel.ch (A.B.); 2Department of Human Genetics, Inselspital Bern, University of Bern, CH-3010 Bern, Switzerland; nijas.aliu@insel.ch; 3Department of Pediatric Surgery, Children’s Hospital of Eastern Switzerland, CH-9000 St. Gallen, Switzerland; frank-martin.haecker@kispisg.ch; 4Faculty of Medicine, University of Basel, CH-4031 Basel, Switzerland

**Keywords:** cytokine, JAK-STAT pathway, Wilms tumor, pediatric kidney tumor, signaling

## Abstract

Renal tumors comprise ~7% of all malignant pediatric tumors. Approximately 90% of pediatric kidney tumors comprise Wilms tumors, and the remaining 10% include clear cell sarcoma of the kidney, malignant rhabdoid tumor of the kidney, renal cell carcinoma and other rare renal tumors. Over the last 30 years, the role of cytokines and their receptors has been considerably investigated in both cancer progression and anti-cancer therapy. However, more effective immunotherapies require the cytokine profiling of each tumor type and comprehensive understanding of tumor biology. In this study, we aimed to investigate the activation of signaling pathways in response to cytokines in three pediatric kidney tumor cell lines, in WT-CLS1 and WT-3ab cells (both are Wilms tumors), and in G-401 cells (a rhabdoid kidney tumor, formerly classified as Wilms tumor). We observed that interferon-alpha (IFN-α) and interferon-gamma (IFN-γ) very strongly induced the activation of the STAT1 protein, whereas IL-6 and IFN-α activated STAT3 and IL-4 activated STAT6 in all examined tumor cell lines. STAT protein activation was examined by flow cytometry and Western blot using phospho-specific anti-STAT antibodies which recognize only activated (phosphorylated) STAT proteins. Nuclear translocation of phospho-STAT proteins upon activation with specific cytokines was furthermore confirmed by immunofluorescence. Our results also showed that both IFN-α and IFN-γ caused upregulation of major histocompatibility complex (MHC) class I proteins, however, these cytokines did not have any effect on the expression of MHC class II proteins. We also observed that pediatric kidney tumor cell lines exhibit the functional expression of an additional cytokine signaling pathway, the tumor necrosis factor (TNF)-α-mediated activation of nuclear factor kappa B (NF-κB). In summary, our data show that human pediatric renal tumor cell lines are responsive to stimulation with various human cytokines and could be used as in vitro models for profiling cytokine signaling pathways.

## 1. Introduction

Renal malignancies are common in children and account for approximately 7% of all pediatric malignant tumors, of which ~90% are Wilms tumors, also known as nephroblastoma, and most often affect children aged 3 to 4. The other 10% include clear cell sarcoma of the kidney, renal cell carcinoma, rhabdoid tumor of the kidney and other rare kidney tumors. Each year, about 950 children in Europe and 500 children in the United States are diagnosed with a kidney tumor. In general, Wilms tumor is a curable disease, the treatment includes surgery, radiation and chemotherapy [1,2,3,4,5,6].

The Janus tyrosine kinase–signal transducer and activator of transcription (JAK–STAT) pathway is involved in many physiological processes including cell differentiation, organ development, cell proliferation, cell cycle, cell survival, cell death and the development of autoimmune diseases [7,8,9]. This pathway is among the highly conserved pathways in a wide range of species. Over 50 cytokines signal via the JAK-STAT pathway and due to their diverse biological functions, aberrations in JAK-STAT signaling might cause a wide variety of detrimental consequences [10,11,12]. Dysregulation of the JAK–STAT pathway very often has been observed in many primary tumors and can lead to angiogenesis. In addition, this pathway is involved in the induction or prevention of apoptosis [13].

The JAK-STAT cytoplasmic protein family functions in signaling and transcription factor regulation, participating in normal cellular responses to cytokines and growth factors. Signaling pathways are activated through cell membrane receptors, which are triggered by an array of ligands [9,14,15]. When these ligands are bound to the transmembrane receptors, it causes rapid phosphorylation and thus activation of members of the intracellular JAK protein family. Cytokines form a very diverse group of signaling proteins which are secreted by a broad range of cell types, including immunocompetent cells, and contribute to angiogenesis, inflammation, cell migration and cell damage, among many other cellular processes. The activated receptor–kinase complexes recruit members of the STAT proteins, which then translocate to the nucleus where they subsequently promote transcription of specific gene [10,11,12].

STATs are a class of transcription factors which are activated upon tyrosine (or serine) phosphorylation and they form homo- or heterodimers that translocate to the cell nucleus where they function as activators of transcription [7,8,9,10,11,12]. There are seven mammalian STAT family members that have been discovered to date, namely STAT1, STAT2, STAT3, STAT4, STAT5a, STAT5b and STAT6, and these STATs share a common structural motif [7,8,10].

The first two STAT proteins were identified in the interferon (IFN) system. STAT1 is activated in response to a large number of ligands and is essential for responsiveness to IFN-α and IFN-γ. Phosphorylation of STAT1 on Tyr701 induces STAT1 dimerization, rapid nuclear translocation and DNA binding [11,14]. As phosphorylation of STAT1 at Tyr701 is essential for dimerization and DNA binding, it is an excellent marker to measure STAT1 activity. STAT1 plays an important role in immune responses against viral, fungal and mycobacterial pathogens. In addition to tyrosine phosphorylation, STAT1 could also be phosphorylated at Ser727 [15,16] and STAT1 is usually in its active form in many tumors.

STAT2 is activated in response to IFN stimulation and plays an important role in immune responses and carcinogenesis [7,10]. Unlike other STAT proteins, STAT2 does not form homodimers, instead activated STAT2 (phosphorylated at Tyr690) forms a heterodimer with STAT1 protein and then translocates to the nucleus. It has also been reported that in B-cells STAT2 is able to form a complex with STAT6 and p48 after activation by IFN-α [17].

STAT3 is considered a key signaling molecule for many cytokines and growth factor receptors. It has been reported that STAT3 is constitutively activated in a number of human tumors [15,18]. A variety of cytokines cause activation of STAT3 protein. Activation of STAT3 is accompanied by tyrosine phosphorylation at Tyr705, which induces dimerization, nuclear translocation and DNA binding. In some cases, STAT3 is also phosphorylated at Ser727 through the mitogen-activated protein kinase (MAPK) or mammalian target of rapamycin (mTOR) pathway [14,16].

Several organs (thymus, testis, spleen) and activated T-cells express high levels of STAT4. Phosphorylation at Tyr693 leads to activation of STAT4 [18]. It is activated in response to interleukin (IL)-12 in natural killer (NK) cells, however, not in T-cells [19].

STAT5a and STAT5b are the two most closely related STAT proteins, which play very important roles in the survival and proliferation of hematopoietic cells. Both proteins are activated in response to a variety of ligands, including IL-2 family cytokines [20,21]. Phosphorylation at Tyr694 is obligatory for STAT5 activation and it has been shown that STAT5 is constitutively activated in several leukemia cell types [22].

Lastly, STAT6 is activated in response to IL-4 and IL-13 [23,24] via phosphorylation at Tyr641 [25] and, in Daudi cells (also called Burkitt lymphoma) after stimulation with IFN-α, STAT6 forms an unusual heterodimer with STAT5 protein [26].

In the human genome, the seven STAT family members described above are organized in three main clusters: genes encoding STAT1 and STAT4 are located on the 2q chromosome, STAT2 and STAT6 on 12q and STAT3, STAT5a and STAT5b genes on 17q [27].

On the other hand, when cytokines are considered, tumor necrosis factor (TNF) is a central cytokine in inflammatory reactions. This cytokine is known to activate the nuclear factor kappa B (NF-κB) as well as MAPK and phosphatidylinositol 3-kinase (PI3K)/protein kinase B (AKT) pathways. Activated T-cells and macrophages are the primary producers of TNF-α in response to inflammation and infection [28,29] and inhibition of TNF-α has proven successful in the treatment of many autoimmune disorders [30].

Transcription factors of the NF-κB/Rel family are crucial for inflammatory and immune responses [31]. There are five family members, RelA, c-Rel, RelB, NF-κB1 (p105/p50) and NF-κB2 (p100/p52), which have been described [31,32]. To form dimeric complexes that bind DNA and regulate transcription, the Rel proteins bind p50 and p52. In unstimulated cells, NF-κB is “kept” in cytoplasm by IκB inhibitory proteins [32]. NF-κB-activating substances can induce the phosphorylation of IκB proteins, targeting them for rapid degradation through the ubiquitin-proteasome pathway and releasing NF-κB to enter the nucleus where it regulates gene expression. In various inflammatory diseases, deregulated NF-κB activation contributes to pathogenic processes [33,34,35].

Lately, the role of cytokines and their vast signaling networks has been extensively investigated in both tumor progression and anti-cancer treatments. However, effective immunotherapies, especially for pediatric tumors, require the cytokine profiling of each tumor type and comprehensive understanding of tumor biology [36,37].

The aim of the present work was to study the interaction of cytokines with pediatric kidney tumor cell lines in view of determining the importance of these processes for signal transduction. Cells included in this study were: the WT-CLS1 and WT-3ab cell lines, derived from Wilms tumor patients, and the G-401 cell line established from a 3-mnoth-old child with a rhabdoid tumor of the kidney. Our goal was to investigate the presence and functionality of both JAK-STAT and the NF-κB pathways upon cytokine stimulation with IL-4, IL-6, IFN-α, IFN-γ and TNF-α.

## 2. Results

To examine the effects of cytokines on the activation of STAT proteins in pediatric kidney tumor cell lines WT-CLS1, WT-3ab and G-401, Western blot, immunofluorescence, MTT and flow cytometry experiments have been performed.

### 2.1. Western Blot Analysis of STAT Proteins

The cells were treated with appropriate cytokines or left untreated as a negative control. As shown in Figure 1A, a strong activation of phopsho-STAT1 protein was observed in all studied cell lines after incubation with both interferon alpha (IFN-α) and interferon gamma (IFN-γ) when the membranes were probed with anti-phospho-STAT1 antibodies (Figure 1A, upper part). Anti-phospho-STAT1 antibody detects endogenous levels of STAT1 only when STAT1 protein is phosphorylated at Tyr701. This antibody does not cross react with corresponding phospho-tyrosines of other STAT proteins. We did not observe any differences between expressions of phospho-STAT1 in WT-CLS1, WT-3ab or G-401 cells. We also probed the membranes with a STAT1 antibody, which detects unphosphorylated STAT1 (Figure 1A, lower part). The activation of STAT1 was specific, since the cytokine-untreated samples did not show any traces of phospho-STAT1 but expressed high amounts of unphosphorylated STAT1 (Figure 1A, lower part). The effects of cytokines on ARPE-19 cells were examined earlier [38] and in the present study served as positive (when the ARPE-19 cells were treated with a specific cytokine) and negative (untreated ARPE-19 cells) controls.

In the next set of experiments, we examined the activation of STAT3 molecules in response to stimulation with IFN-α and IL-6. As shown in Figure 1B (upper part), a strong band corresponding to phospho-STAT3 protein after stimulation with both IFN-α and IL-6 was detected, and the samples which were left untreated did not show any phospho-STAT3 (Figure 1B). In contrast, when the membranes were probed with STAT3 antibody, which recognizes total STAT3 protein independent of phosphorylation, we observed the band corresponding to STAT3 in all samples (Figure 1B, lower panel). Phospho-STAT3 antibody recognizes STAT3 protein only when it is phosphorylated at Tyr705. Finally, the cells were also treated with human recombinant IL-4 and a strong band of phospho-STAT6 was observed (Figure 1C, upper part). When the samples were probed with STAT6 antibody, which recognizes total STAT6, a strong band was detected in all samples (Figure 1C, lower part).

### 2.2. Immunofluorescence Analysis of Phosphorylated STAT Proteins

The nuclear translocation of STAT proteins was visualized using immunofluorescence microscopy. As shown in Figure 2 (right panel), when the cells were treated with IFN-γ and stained with anti-phospho-STAT1 antibody a strong nuclear translocation was detected. The cytoplasm was stained with anti-vinculin antibody (green). In contrast, in all examined unstimulated cells (Figure 2, left panel) only cytoplasmic vinculin staining was visible. The cells were also stimulated with IFN-α, IL-6 and IL-4. The nuclear translocation was observed only in stimulated cells whereas in unstimulated cells only vinculin staining was visible and no phospho-STAT staining was detected in the nuclei (Appendix A).

### 2.3. Flow Cytometry Analysis of STAT Proteins and MHC Modulation

Response to cytokines in three pediatric kidney tumor cell lines was also analyzed by flow cytometry. Cells were incubated for 20 min with appropriate cytokine and analyzed for expression of phospho-STAT proteins. Expression of phosphorylated STAT1 after stimulation with IFN-α and IFN-γ is shown in Figure 3A (left part). The WT-CLS1 cells expressed higher amounts of phospho-STAT1 after stimulation with IFN-γ (blue histogram) compared to stimulation with IFN-α (red histogram).

Expression of phospho-STAT3 (Figure 3A, middle part) was observed after stimulation with both IFN-α (red histogram) and IL-6 (green histogram). We also examined whether IL-4 is able to activate STAT6 in WT-CLS1 cells. As shown in Figure 3A (right part), stimulation with IL-4 caused strong activation of phospho-STAT6 protein (orange histogram). Very similar results were obtained when we analyzed the activation of STAT proteins in two other cell lines, WT-3ab and G-401 (Appendix A, left part).

Responsiveness to IFN-α and IFN-γ treatment was also observed in MHC modulation experiments. All studied pediatric kidney tumor cells express MHC class I as shown in Figure 3B (grey histogram) (and Appendix A for WT-3ab cells and S2D for G-401 cells), compared to isotype controls (grey dotted histograms). Incubation with IFN-α and IFN-γ led to the overexpression of MHC class I, whereas response to IFN-γ was much stronger (Figure 3B, left part). The ratio of mean fluorescence intensity of the IFN-α peak to the mean of fluorescence intensity of unstimulated cells was 3.92 and the ratio of fluorescence intensity of the peak corresponding to IFN-γ to unstimulated cells was 5.45.

The studied cells do not express MHC class II and incubation with IFN-α and IFN-γ did not change the MHC class II expression pattern, and all histograms are almost overlapped (Figure 3B, right part, Appendix A, right part).

### 2.4. Effect of Cytokines in MTT Assay

To evaluate the role of cytokines on the proliferative capacity of kidney tumor cells, we further performed an MTT assay. As shown in Figure 4, both IFN-α and IFN-γ suppressed proliferation of WT-CLS1 cells whereas both IL-4 and IL-6 stimulated their proliferation. MTT assay has shown that IFN-α significantly suppressed the metabolic activity of WT-CLS1 cells. In contrast, the treatment of cells with IL-4 led to increasing proliferation activity. Similar results were obtained for WT-3ab and G-401 cells (Appendix A).

### 2.5. Karyotyping Analysis of Pediatric Kidney Tumor Cells

The cytogenetic analyses have revealed that all three studied cell lines show different karyotypes. WT-CLS1 cells have the 46,XX karyotype without visible chromosomal aberrations (Appendix A). In contrast, the WT-3ab cell line had multiple numerical and structural aberrations and the G-401 cell line has a sole structural aberration in chromosome 12 (Appendix A).

### 2.6. Nuclear Factor Kappa B Signaling Pathway Activation in Pediatric Kidney Tumor Cells

We observed that another cytokine signal transduction pathway could be activated in WT-CLS1, WT-3ab and G-401 cells, namely stimulation of cells with human TNF-α leads to activation and nuclear translocation of the NF-κB/p65 transcription factor complex. As shown in Figure 5, all examined samples constitutively express p65, but its expression increased after TNF-α stimulation. In contrast, when the membranes were incubated with NF-κB antibody recognizing total p65 independent of phosphorylation we observed a strong band of p65 in all samples. The lysate of ARPE-19 cells treated with human recombinant TNF-α was used as a positive control.

## 3. Discussion

Several biochemical methods were applied in this study to examine the activation of signaling pathways in response to human recombinant cytokines. All examined STAT family members (STAT1, STAT3 and STAT6) could be activated by a short treatment (20 min) using corresponding cytokines. Here, we have investigated the activation of the JAK-STAT signaling pathway in three pediatric kidney tumor cell lines (WT-CLS1, WT-3ab and G-401) and demonstrated that several STAT proteins are activated in these cell lines in response to IFN-α, IFN-γ, IL-6 and IL-4. Our data indicate that IFN-γ very strongly induced the activation of STAT1 in all applied assays (Western blot, immunofluorescence and flow cytometry, Figure 1, Figure 2 and Figure 3). IFN-α induced less strong activation of STAT proteins (both STAT1 and STAT3) compared to IFN-γ and IL-6 (Figure 2 and Figure 3). The individual activation patterns were cytokine-specific, and major results are summarized in Table 1 and in a scheme (Appendix A).

An additional signaling pathway (NF-κB) was also activated in all examined cell lines upon stimulation with human TNF-α. Very useful tools for our studies were phospho-specific antibodies, which recognize only phosphorylated STAT or NF-κB proteins. These antibodies were applied in our Western blot experiments (Figure 1 and Figure 5). Using phospho-STAT antibodies, we observed a nuclear translocation of examined STAT proteins (Figure 2 and Appendix A) after stimulation with an appropriate cytokine in immunofluorescence experiments. We have tested several phospho-NF-κB antibodies for immunofluorescence assay and for flow cytometry, but all these antibodies were unspecific.

The cytogenetic studies have demonstrated that the WT-CLS1 cell line has a normal 46,XX karyotype, in contrast to WT-3ab cells, in which multiple numerical and structural aberrations were detected (Appendix A), which are in line with results described by C. Stock [39]. The G-401 cell line has a sole structural aberration in chromosome 12 arising from partial trisomy 7p, and this chromosomal aberration was described in detail earlier [40]. Interestingly, in all three examined pediatric kidney tumor cell lines, incubation with human cytokines led to the activation of both JAK-STAT and NF-κB signaling pathways independent of their karyotype. In all cell lines, we observed very similar results (Figure 1, Figure 2, Figure 3 and Figure 4 and Appendix A).

In the last several years, the immunotherapy approach has become a key therapeutic strategy in the treatment of many hematologic malignancies and adult solid tumors. One mechanism by which a tumor can avoid immunosurveillance is the modulation (downregulation) of MHC proteins. It has been described that the MHC protein family is responsive to modulation by inflammatory cytokines [41,42]. In our present study, IFN-α and IFN-γ stimulation caused the upregulation of MHC class I proteins (Figure 3) and did not have any effect on the expression of MHC class II in any tested cells.

Interferons (IFNs) have traditionally been used for cancer treatment due to their anti-tumor effects. But IFNs are also involved in promoting the outgrowth of tumor cells. Significant research efforts are required to investigate molecular mechanisms of their pro- and anti-tumorigenic effects. New IFN-based strategies should be developed, as IFNs appear to play a crucial role in immunotherapy responses [43,44].

Recent advances in immune-oncology have shown the benefits of targeting the immune system for the treatment of various cancers. Immunotherapies, such as immune checkpoint inhibitors, have improved the prognosis and survival of many cancer patients, but despite this success, immunotherapy for pediatric solid tumors remains in the early stages of development [45,46].

Currently, multiple drugs have been developed to target the JAK-STAT pathway. These drugs could be divided into three subtypes: cytokine or receptor antibodies and JAK and STAT inhibitors. Moreover, novel agents also continue to be developed and tested in preclinical and clinical studies [47,48]. The JAK-STAT pathway is a potential target for treatment of pediatric kidney tumors and research efforts have been aimed at understanding mechanisms of resistance to immunotherapy and how anti-tumor immune response can be therapeutically enhanced. It has been shown that tumor cell recognition by the immune system plays a key role in effective response to T-cell-targeting therapies in patients.

Taken together, our data have shown that IFN-α activated STAT1 and STAT3 in all examined cell lines, whereas IL-6 and IL-4 activated STAT3 and STAT6, respectively. The activation of STAT1 after incubation with human recombinant IFN-α was relatively strong in Western blot but much weaker in immunofluorescence and flow cytometry assays. It should be kept in mind that different biochemical methods must be applied to characterize the activation of STAT proteins.

Data on the activation of JAK-STAT signaling pathway in pediatric kidney cells are very limited. Therefore, we cannot directly compare our findings to those of the current literature. More fundamental research is needed to determine the specific role the pathway plays in individual disease.

## 4. Materials and Methods

### 4.1. Materials

DMEM/F12, RPMI 1640, Iscove’s and McCoy’s 5a media for cell culture, fetal calf serum (FCS), trypsin-EDTA and cell culture supplements were purchased from Bioconcept (Allschwil, Switzerland). Human recombinant interferon alpha (IFN-α), interferon gamma (IFN-γ), interleukin (IL)-6, IL-4 and human recombinant tumor necrosis factor alpha (TNF-α) were from R&D Systems (Minneapolis, MA, USA). Primary rabbit polyclonal antibodies against phosphorylated and unphosphorylated human STAT proteins for Western blot experiments were from Cell Signaling (Danvers, MA, USA). Recognized phosphorylated tyrosines on the respective STAT proteins can be summarized as follows: epitope Y701 on STAT1, epitope Y705 on STAT3 and epitope Y641 on STAT6. Primary rabbit antibodies against phosphorylated STAT proteins were also used for immunofluorescence experiments. Rabbit monoclonal antibody recognizing human phosphorylated nuclear factor kappa B (NF-κB) protein was from Cell Signaling (Danvers, MA, USA) and antibody recognizing unphosphorylated NF-κB (p65) was from Santa Cruz (Dallas, TX, USA). Monoclonal antibodies against human vimentin for immunofluorescence experiments were from Santa Cruz (Dallas, TX, USA). Secondary antibodies conjugated with Alexa Fluor 488 (goat anti-mouse) and Alexa Fluor 555 (goat anti-rabbit) were from Merck (Buchs, Switzerland). Antibodies for flow cytometry used in this study were: Alexa Fluor 488 mouse anti-human phospho-STAT1 (epitope Y701), anti-phospho-STAT3 (epitope Y705), anti-phospho-STAT6 (epitope Y641), peridinin–chlorophyll–protein complex (PerCP)-conjugated mouse anti-human monoclonal antibodies against major histocompatibility complex (MHC) class I human leukocyte antigen-A,B,C (HLA-A,B,C), FITC-conjugated mouse anti-human HLA-DR and HLA-DQ, as well as isotype-matched IgG for flow cytometry experiments, were purchased from BD Biosciences (San Jose, CA, USA). All other chemicals employed in this study were from Merck (Buchs, Switzerland) and of the highest grade of purity. All the cell culture experiments were performed in TPP plastic ware (Trasadingen, Switzerland).

### 4.2. Cell Culture

Human retinal pigment epithelial cell line ARPE-19 (CRL-2302) and human pediatric rhabdoid tumor (formerly classified as Wilms Tumor, CRL-1441) were purchased from the American Type Culture Collection (ATCC). Human Wilms tumor cell line WT-CLS1 was purchased from Cell Lines Services (CLS GmBH, Eppelheim, Germany). All cell lines were accompanied by identification test certificates and were grown according to corresponding tissue culture collection. Wilms tumor cell line WT-3ab was kindly provided by Dr. C. Stock (Vienna, Austria). The ARPE-19 cells were grown in Dulbecco’s minimal essential medium (DMEM) DMEM/F12, supplemented with 10% FCS and 1% kanamycin solution at 37 °C and 5% CO_2_. The G-401 cells were grown in McCoy’s 5a and WT-3ab cells in RPMI 1640 medium supplemented with 10% FCS and 1% kanamycin solution. The WT-CLS1 cells were cultivated in Iscove’s medium, supplemented with 2mM L-glutamine and 10% FCS.

### 4.3. Western Blot

Western blot analysis of phospho-STAT and unphosphorylated STAT protein was performed as described before [49]. Briefly, cells were treated with the appropriate cytokine (10 ng/mL) for 20 min at 37 °C or left untreated (negative control) and then were lysed in a lysis buffer (50 mM Tris–HCl, 5 mM EDTA, 150 mM NaCl, 0.5% Nonidet-40, 1 mM PMSF, 10 µM sodium vanadate and protein inhibitors aprotinin, leupeptin and pepstatin (1µg/mL each), pH 8.0) on ice for 30 min. After centrifugation for 5 min at full speed, 40 µg total protein was mixed with 4× NuPAGE LDS loading buffer from Invitrogen (Carlsbad, CA, USA) and resolved on NuPAGENovex 4–12% Bis-Tris gels. The proteins were transferred to a nitrocellulose membrane using a Novex Tris-Glycine Transfer buffer (Invitrogen, Carlsbad, CA, USA) according to the manufacturer’s instructions. Nonspecific binding sites were blocked with 5% milk in TBST (120 mM Tris–HCl, pH 7.4, 150 mM NaCl and 0.05% Tween 20) for 1 h at room temperature. Target proteins were detected using specific STAT antibodies recognizing phospho-STAT proteins or total STAT proteins, see Section 4.1. The membranes were washed three times and incubated with secondary antibodies conjugated with horseradish peroxidase. Immune complexes were visualized using an enhanced chemiluminescence system (Bio-Rad, Hercules, CA, USA). Data are representative of three independent experiments with nearly identical results. Whole cell extracts from ARPE-19 cells served as a control [38].

### 4.4. Immunofluorescence Staining

Thirty thousand cells were cultured overnight on 12 mm glass cover slips in 24-well plates. Next day, the cells were incubated with appropriate cytokines (10 ng/mL) for 20 min at 37 °C or left untreated as a negative control and then fixed/permeabilized in ice-cold methanol/acetone (1:1) for 20 min, washed with PBS and further incubated with 10% goat serum in PBS for 1 h at room temperature. Rabbit anti-human phospho-STAT and mouse anti-human vimentin antibodies were used as primary antibodies (dilution 1:100). Goat anti-mouse Alexa Fluor 488 and goat anti-rabbit Alexa Fluor 555 were used as secondary antibodies (dilution 1:1000). The images were collected and analyzed on an Olympus BX-51 microscope with 40× objective using proprietary software as described before [50].

### 4.5. Analysis of Activated STAT Proteins by Flow Cytometry

Flow cytometry experiments were performed as previously described [38]. Briefly, cells were stimulated with 10 ng/mL cytokine for 20 min at 37 °C, washed, harvested and then fixed with 2% PFA for 30 min at RT and permeabilized with 0.1% TX-100 for 10 min at RT. Then, unstimulated and cytokine-stimulated cells were stained with Alexa-Fluor-488-conjugated anti-phospho-STAT1, anti-phospho-STAT3 or anti-phospho-STAT6 antibodies as indicated in figure legends and measured on the FACS Calibur cytometer with Cell Quest Pro software (Becton Dickinson, Franklin Lakes, NJ, USA). Unstained cells or cells stained with isotype-matched IgG antibodies served as a negative control. All experiments were performed three times. All staining data are shown as histograms or expressed as median fluorescence intensity (MFI) values, calculated as the ratio between the experimental MFI and that observed upon staining with an isotype-matched control antibody (50,000 events were analyzed in each variant).

### 4.6. MHC Class I and Class II Modulation Assay

Surface expression of MHC class I and class II proteins was monitored by flow cytometry, using a PerCP-labeled mouse anti-human monoclonal antibody for HLA-A,B,C heavy chain and fluorescein isothiocyanate (FITC)-conjugated mouse anti-human antibodies against MHC class II (HLA-DR, HLA-DQ) or control, isotype-matched reagent in cells cultured for 48 h in the presence or absence of IFN-α and IFN-γ as described previously [50]. Mean fluorescence intensity of stained cells was measured and analyzed using a FACS Calibur cytometer and the Cell Quest Pro software, v5.1.

### 4.7. MTT Assay

Thirty thousand cells were placed in 24-well cell culture plates, and the next day the cytokines were added into wells (10 ng/mL) and incubated for 96 h. Then, 0.1 mg/mL MTT was added to the wells and the cells were incubated for a further 4 h. The reaction was stopped by adding 125 µL of DMSO. The supernatants were harvested and the optical density was measured at 590 nm as described previously [51]. Four independent experiments were performed in triplicate.

### 4.8. Chromosome Analysis

WT-CLS1, WT-3ab and G-401 cells were cultivated in well plates in appropriate medium. Next day, the cells were incubated with colcemid and metaphases for chromosome spread were observed under a phase contrast microscope as previously described [52]. At least 50 metaphases were analyzed by an experienced cytogeneticist. For image acquisition and karyotyping, the proprietary Genikon program was used.

## 5. Conclusions

In this study, the activation of JAK-STAT and NF-κB signaling pathways in three pediatric kidney tumor cell lines, WT-CLS1, WT-3ab and G-401, in response to various cytokines was investigated using several biochemical methods. Our data showed that STAT1, STAT3 and STAT6 are activated upon stimulation with various human cytokines, and TNF-α induced a strong activation of the NF-κB pathway in these cells. With the help of phospho-specific antibodies recognizing only phosphorylated STATs, we observed a translocation of STAT proteins into the nuclei of cytokine-stimulated kidney tumor cells. Moreover, both IFN-α and IFN-γ caused upregulation of MHC class I in all examined cell lines. The JAK-STAT signaling pathway might have clinical potential for diagnosis and therapy of Wilms tumor patients with poor prognosis. Our data demonstrated that the examined human pediatric kidney tumor cell lines are responsive to stimulation by various cytokines and could be used as in vitro models for profiling cytokine signaling pathways.

## Figures and Tables

**Figure 1 ijms-25-02281-f001:**
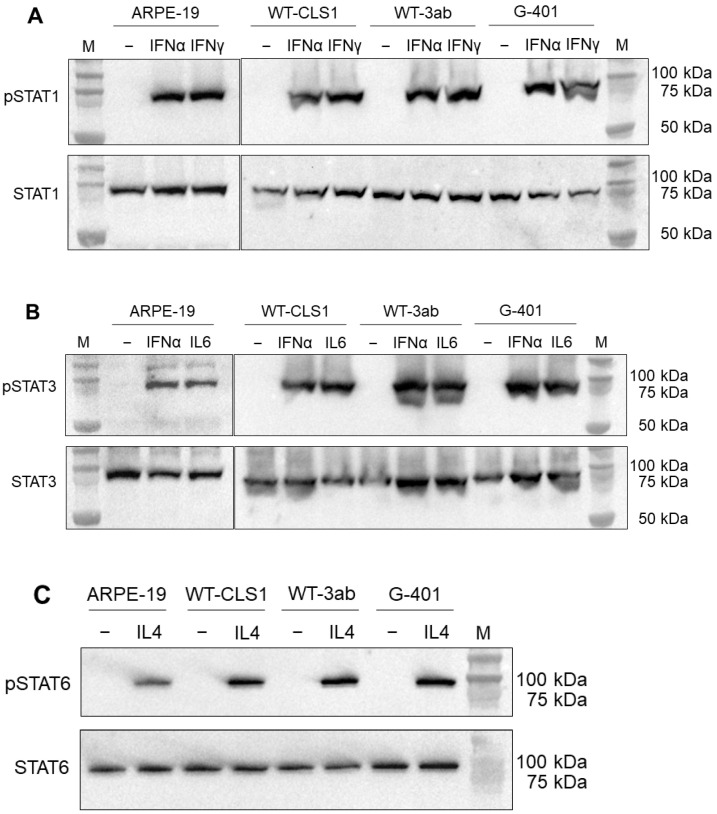
The ARPE-19, WT-CLS1, WT-3ab and G-401 cells were treated with 10 ng/mL of cytokine for 20 min. The membranes were probed with phospho-specific antibodies recognizing only translocated STAT proteins (see Section 4); (**A**)—cells were stimulated with IFN-α and IFN-γ and probed with anti-phospho-STAT antibody; (**B**)—cells were stimulated with IFN- α and IL-6 and probed with anti-phospho-STAT3 antibody; (**C**)—cells were stimulated with IL-4 and probed with anti-phospho-STAT6 antibody. In addition, the membranes were probed with corresponding STAT antibodies, which recognize total STAT protein independent of phosphorylation (**A**–**C**—lower part). As a positive control, the lysates of ARPE-19 cells treated with appropriated cytokines were used. M—molecular weight marker.

**Figure 2 ijms-25-02281-f002:**
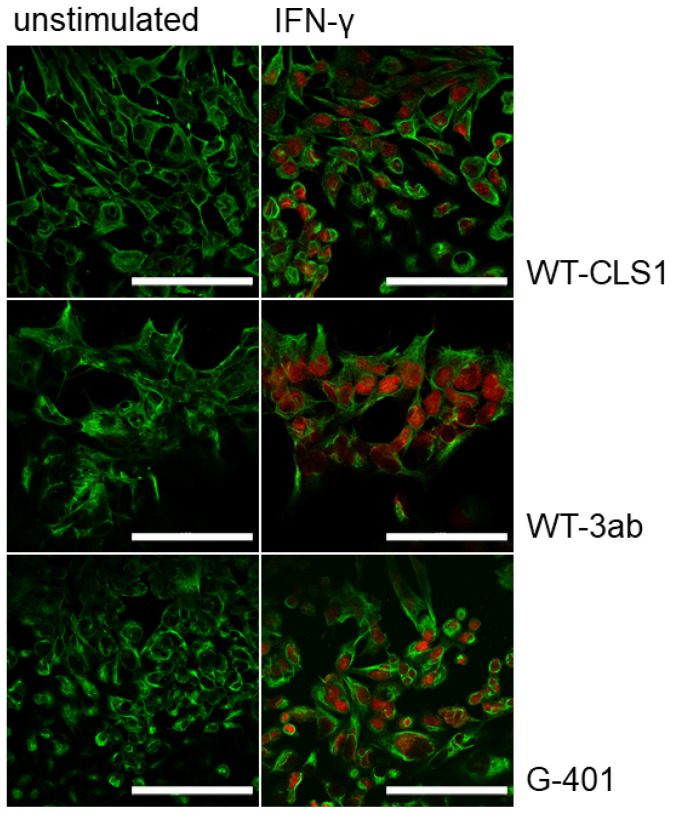
Immunofluorescence analysis of phospho-STAT1 protein in WT-CLS1, WT3ab and G-401 cells. Untreated (**left panel**) and IFN-γ-treated (**right panel**) cells were stained with anti-phospho-STAT1 and anti-vinculin antibodies. An intense nuclear immunostaining of phosphor-STAT1 was observed (**right panels**). The cytoplasmic vinculin staining is shown in green. In all images, magnification 60×, scale bar 50 μm. Images are from one experiment representative of three which gave similar results.

**Figure 3 ijms-25-02281-f003:**
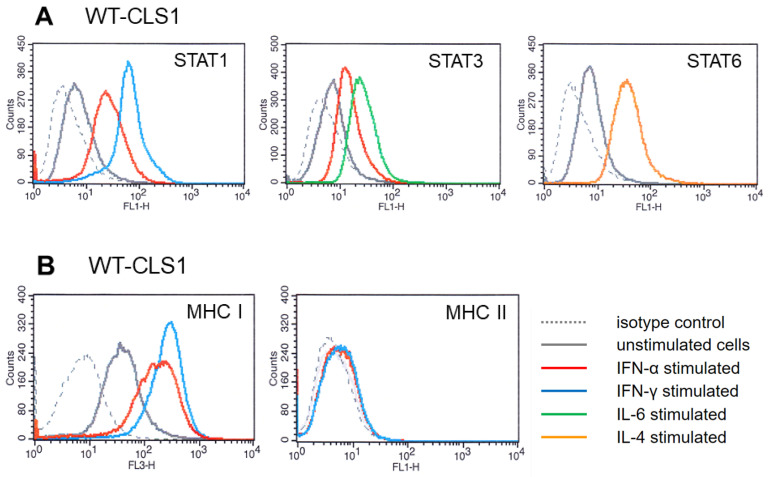
Flow cytometry analysis of WT-CLS1 cells stimulated with IFN-α, IFN-γ, IL-6 as well as with IL-4 and probed with anti-phospho-STAT antibodies. (**A**) (left part, blue histogram): Phospho-STAT1 staining. Fluorescence intensity of WT-CLS1 cells stained with isotype-matched control antibodies (negative control)—grey dotted histogram, unstimulated cells probed with anti-phospho-STAT1 antibody—grey histogram, WT-CLS1 cells stimulated with IFN-α (red histogram) and stimulated with IFN-γ (blue histogram). Middle part: After stimulation with IFN-α and IL-6 and analysis of phospho-STAT3. Grey dotted histogram—isotype-matched negative control. Unstimulated cells—grey histogram, cells stimulated with IFN-α (red histogram) and stimulated with IL-6 (green histogram). Right part: After stimulation with IL-4 and analysis of phospho-STAT6. Grey dotted histogram—isotype-matched control. Unstimulated cells—grey histogram, stimulated with IL-6 (orange histogram). (**B**)—MHC class I and class II modulation. Left part: Flow cytometry analysis of WT-CLS1 cells stimulated with IFN-α and with IFN-γ for 48 h. The WT-CLS1 cells express high amounts of MHC class I (grey histogram), after stimulation with IFN-α the overexpression of MHC class I was observed (red histogram) and with IFN-γ (blue histogram). The grey dotted histograms represent isotype-matched negative controls. Right: MHC class II modulation. Flow cytometry analysis of WT-CLS1 cells stimulated with IFN-α and IFN-γ. The grey dotted histogram shows isotype-matched negative control. WT-CLS1 cells do not express MHC class II (grey histogram). Stimulation with IFN-α (red histogram) and IFN-γ (blue histogram) did not influence the MHC class II expression in WT-CLS1 cells. All four histograms overlap with each other.

**Figure 4 ijms-25-02281-f004:**
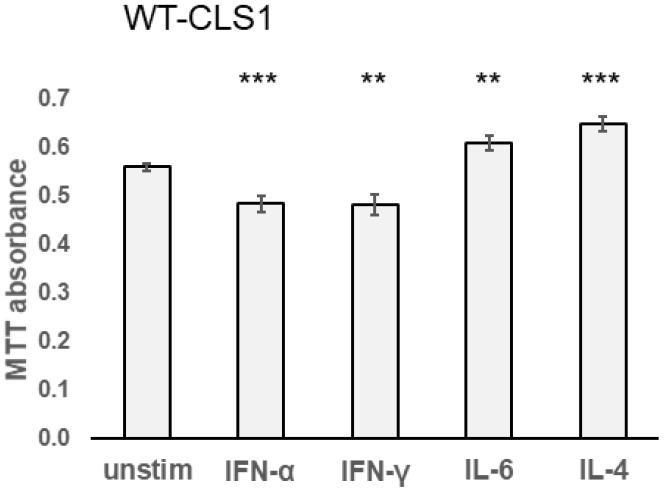
WT-CLS1 cells were incubated with various cytokines (96 h) as outlined in Section 4. IFN-α as well as IFN-γ suppressed but both IL-6 and IL-4 increased the metabolic activity of WT-CLS1. Values represent mean ± SD. Three independent experiments were performed in triplicate. Statistical probabilities (*p*) are expressed as ** when *p* < 0.01 and *** when *p* < 0.001.

**Figure 5 ijms-25-02281-f005:**
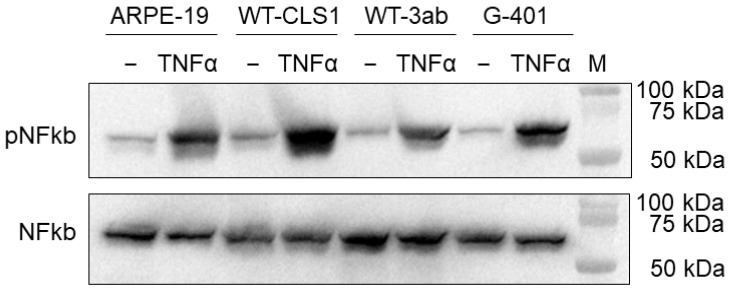
NF-κB signaling in WT-CLS1, WT-3ab and G-401 cells. Cells were treated with human TNF-α as outlined in Section 4. The membranes were probed with anti-phospho-p65 recognizing only phosphorylated p65 and with corresponding NF-κB/p65 antibody, which recognizes total p65 independent of phosphorylation.

**Table 1 ijms-25-02281-t001:** Summary of cytokine-specific activation patterns in WT-CLS1, WT-3ab and G-401 cells.

Cytokine	Protein	WB	IF	FC
IFN-α	STAT1	+++	++	++
IFN-γ	STAT1	+++	+++	+++
IFN-α	STAT3	+++	++	+
IL-6	STAT3	+++	++	++
IL-4	STAT6	+++	++	++

Notes: +++ = strong expression, ++ = moderate expression, + = weak expression. Abbreviations: WB—Western blot, IF—immunofluorescence staining, FC—flow cytometry.

## Data Availability

Data are contained within the article.

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
