# Peer review of "Cytokine Signaling in Pediatric Kidney Tumor Cell Lines WT-CLS1, WT-3ab and G-401"

_ijms, 2024, doi:10.3390/ijms25042281_

Round 1

Reviewer 1 Report

Comments and Suggestions for Authors

Overall, a very sound study demonstrating the uniform pattern of proinflammatory pathways response pattern to cytokines across several paediatric kidney tumour cell lines.

Well executed, well described. All techniques are appropriate and correctly done.

I have only 3 minor corrections/comments:

Line 248 Figure 4 legend – could you please indicate after what time was MTT done. It is written in section 4.7 methods (96h), but it would be useful in the figure legend as well for readers convenience.

Line 123 “However, effective immunotherapies especially for pediatric tumours require the cytokine profiling of each tumour type and comprehensive understanding of tumour biology [36, 37]”. Could you please give 1-2 examples of immunotherapies used or proposed for that Wilms tumours.

Line 335 “Data on the activation of JAK-STAT signaling pathway in pediatric kidney cells are

unfrequent”. Could you please rephrase. 

Author Response

Dear Editor,

Thank you for considering our manuscript for resubmission. Many thanks to all three reviewers for taking the time to review our work and for your valuable comments, which we have now fully addressed. All changes are highlighted in yellow.

We have prepared a point-by-point response to the Referees’ comments, addressing each point raised (s. below).

We hope that the revised manuscript is improved to such a level that it might be accepted for publication in your journal.

Thank you once again for your consideration.

Yours sincerely,

Elizaveta Fasler-Kan and Steffen Berger on behalf of all the Authors

Reviewer 1

  1. Comment: Overall, a very sound study demonstrating the uniform pattern of proinflammatory pathways response pattern to cytokines across several paediatric kidney tumour cell lines

Well executed, well described. All techniques are appropriate and correctly done

Response. Thanks!

  1. Comment: Line 248 Figure 4 legend – could you please indicate after what time was MTT done. It is written in section 4.7 methods (96h), but it would be useful in the figure legend as well for readers convenience

Response Thanks for this comment. Done: please see L 246 in the manuscript

  1. Comment: Line 123 “However, effective immunotherapies especially for pediatric tumours require the cytokine profiling of each tumour type and comprehensive understanding of tumour biology [36, 37]”. Could you please give 1-2 examples of immunotherapies used or proposed for that Wilms tumours

Response. Current treatment of WT patients includes chemotherapy, radiotherapy and surgery.

At present a targeted therapy and immunotherapy are rarely used in the treatment of Wilms tumour patients except in clinical trials. There are three types of study in Wilms tumour immunotherapy, which are inhibition of the COX-2 pathway, chimeric antigen receptor (CAR)-T cell therapy, and multi-tumour associated antigen (TAA)-specific cytotoxic T lymphocytes (CTL) therapy. Among them, the phase I clinical trial of multi-TAA-specific CTL (MTAA-CTL) therapy, which has been completed and the results of this study are very promising.  However, the precise effects of immunotherapy in WT remain to be explore.

  1. Comment: Data on the activation of JAK-STAT signaling pathway in pediatric kidney cells are unfrequent. Could you please rephrase

Response. Thanks for this comments. We have rephrased this sentence to: “Data on the activation of JAK-STAT signaling pathway in pediatric kidney cells are very limited. Please see L 341.

Reviewer 2 Report

Comments and Suggestions for Authors

The manuscript presents a comprehensive study on cytokine signaling in pediatric kidney tumor cell lines, focusing on Wilms tumors and a rhabdoid kidney tumor. The research is highly relevant to the field of pediatric oncology and immunotherapy, offering insights into the activation of signaling pathways in response to cytokines.

Major Comments:

1.       The manuscript is well-structured and presents complex biological processes in an understandable manner. However, some sections could benefit from additional context or explanations, particularly for readers less familiar with cytokine signaling.

2.       The graphical representations effectively aid in understanding the study's findings. It might be beneficial to include schematic diagrams summarizing the signaling pathways activated in each cell line for enhanced clarity.

3.       To further validate the role of IFN-alpha, it is recommended to perform experiments involving the depletion of the IFNAR receptor in one of the cell lines. This could provide stronger evidence for the specific role of IFN-alpha in these pathways.

4.       A detailed description or representation of the gating strategy would enhance the reproducibility and clarity of the flow cytometry results.

5.       Providing quantitative data for the fluorescence images, perhaps using software like QPath, would add significant value to the results.

Author Response

Point-by-point responses to the comments raised by the Referees

Reviewer 2

  1. Comment. The manuscript is well-structured and presents complex biological processes in an understandable manner. However, some sections could benefit from additional context or explanations, particularly for readers less familiar with cytokine signaling

Response.  Thanks for your comment. The JAK-STAT signaling pathway was discovered over 40 years ago and there are many excellent reviews describing the significance of this pathway for health and diseases. We have provided references to several very important reviews for those who would like to learn more about this pathway (References 9, 12, 13, 28, 36, 43). As you have suggested we included more information about the application of interferons for cancer treatment, please see L314-318. We also included additional references.

  1. Comment. The graphical representations effectively aid in understanding the study's findings. It might be beneficial to include schematic diagrams summarizing the signaling pathways activated in each cell line for enhanced clarity

Response. Thanks for pointing this out. As you have suggested we have prepared a graphical representation of our major results. Since all three studied cell lines have shown very similar results, we have prepared one scheme. Please see Supplementary Figure 5 and information about it in a text, L 282-283

  1. Comment. To further validate the role of IFN-alpha, it is recommended to perform experiments involving the depletion of the IFNAR receptor in one of the cell lines. This could provide stronger evidence for the specific role of IFN-alpha in these pathways

Response: Thanks for your comment. IFN-alpha is a very important agent, which is capable of activating all seven STAT proteins depending of the source of the cells. To study the mechanisms of the ligand-inducible elimination of IFNAR1along with its potential for medical application is a very challenging task. We might think about it for the future. Other cytokine receptors, for example IFNGR or IL-6 receptor might be also very important for the studies of pediatric tumours. From our point of view, it would be a separate study. Thanks for this idea!

  1. Comment. A detailed description or representation of the gating strategy would enhance the reproducibility and clarity of the flow cytometry results

Response. Thanks for this comment. As outlined in M&M section all flow cytometry experiments have been performed on a FACS Calibur. You are right, when the different subpopulations of cells in a clinical sample are used in the flow cytometry experiments, then the gating strategy is very important. Gating is usually performed manually. However, there can be considerable disagreement in how gates should be applied, even between individuals experienced in the field. Fortunately, in our study we have used commercially available, homogenous cell lines and could easily analyse 50000 events for each variant. One colour staining have been performed, the antibody were labelled either with Alexa A-488 or with PerCP). When Alexa 488 labeled antibodies were used, then the images were collected in FL1 channel and in MHC modulation experiments a PerCP labelled antibody was used and the data were collected on the Fl3 channel All flow cytometry experiments were performed at least three times and obtained data have shown very similar results.  As you have suggested, we included more information about flow cytometry experiments into M&M section, please see L 424-426.

  1. Comment. Providing quantitative data for the fluorescence images, perhaps using software like QPath, would add significant value to the results

Response. Thanks for your comment. As outlined in M&M section, for our imuunofluorescence experiments we used a fluorescent Olympus BX-51 microscope in the department. All images were collected and analysed using a proprietary software. The goal of the study was to demonstrate the nuclear localization of a specific STAT protein after stimulation with an appropriate cytokine compared to unstimulated cells. We were not interested to analyse the quantitative differences between cytokine stimulated and unstimulated cells nor between stimulation with various cytokines. A “yes or no” translocation answer without a quantification of the fluorescence intensity was sufficient at this stage. Thanks for your suggestion, we might use the QuPath in the future, if necessary

Reviewer 3 Report

Comments and Suggestions for Authors

The rationale of this study should be not only investigating in vitro cytokine profiling signaling pathways in pediatric kidney cancer cells but also evaluating whether cytokines can be candidates for therapeutic agents against pediatric kidney cancer. One of the most effective agents in the treatment of kidney cancer (Renal cell carcinoma) is IL2 cytokine. The authors should have tested IL2 in the pediatric kidney cancer cells. 

Fig 1 and Figure S1: They could be combined to present simply. 

Fig 2 and Figure S2: They could be combined to present simply. The legends of histogram colors could be in the figure so as not to be confused. 

Fig 2 and Figure S3: They could be combined to present simply. 

Author Response

Point-by-point responses to the comments raised by the Referees

Reviewer 3

Comment. The rationale of this study should be not only investigating in vitro cytokine profiling signaling pathways in pediatric kidney cancer cells but also evaluating whether cytokines can be candidates for therapeutic agents against pediatric kidney cancer. One of the most effective agents in the treatment of kidney cancer (Renal cell carcinoma) is IL2 cytokine. The authors should have tested IL2 in the pediatric kidney cancer cells

Response. Thank you very much for this comment. In our study we also investigated the effect of the Il-2 on the activation of STAT5 in these three cell lines. Unfortunately, we did not find good (specific) anti-phospho–STAT5 antibody at least for one biochemical method used. All tested antibodies were very unspecific. If you know a good anti-phospho STAT5 antibody, please let us know. We will be happy to test these antibodies in our further experiments.

Yes, the high dose Il-2 therapy is widely used for treatment of renal cell carcinoma and melanoma patients. However many RC patients do not respond well to chemotherapy. In contrast, many Wilms tumour (WT) patients respond well to a chemotherapy. Mean age of patients suffering from Wilms Tumour is 2.5 years. It is very hard to recruit young children for a clinical trial. Current treatment of WT patients includes chemotherapy, radiotherapy and surgery. Published pediatric data suggest that clinical, molecular and histological characteristics of pediatric RCC differ from adult RCC and from Wilms tumour. More research is necessary in this field.

Comment: Fig 1 and Figure S1: They could be combined to present simply

Fig 2 and Figure S2: They could be combined to present simply. The legends of histogram colors could be in the figure so as not to be confused.

Fig 2 and Figure S3: They could be combined to present simply

Response. Thanks for this comment. In all our previous drafts we put all our experimental data into our main text, that means we combined Figure 1 with Figure S1, Figure 2 with Figure S2 etc.  We observed that the figures were really huge. Each figure contains several subparts. For example Western blot figure 1 has 3 subparts (A.B, C) and demonstrates the data on STAT1, STAT3 and STAT6 activation.  Four cytokines (interferon alpha and interferon gamma, as well as IL6 and IL4) were applied. What is more, we used a phospho-STAT antibody recognizing only phosphorylated STAT protein and also a regular antibody which recognizes total STAT proteins independent of their phosphorylation site. For this reason we have decided to include into main text only experimental data with one of the three examined cell lines, namely WT-CLS and all results with other two cell lines WT-3ab and G-401transfer to a supplementary material. Otherwise, each figure would be three times bigger and it would restrict a logical and smooth reading and understanding of our results.  Please take into consideration that all data obtained from the studied cell lines were very-very similar to each other and from our point of view it was not necessary to show very similar results three times.

Comment: The legends of histogram colors could be in the figure so as not to be confused.

Thank very much for this very valuable comment. As you have suggested, we have prepared a coloured figure legend, which helps to follow the results. Please see our updated FACS figures.
